# Homozygous Splice Site Mutation in *ZP1* Causes Familial Oocyte Maturation Defect

**DOI:** 10.3390/genes11040382

**Published:** 2020-04-01

**Authors:** Özlem Okutman, Cem Demirel, Firat Tülek, Veronique Pfister, Umut Büyük, Jean Muller, Nicolas Charlet-Berguerand, Stéphane Viville

**Affiliations:** 1Institut de Parasitologie et Pathologie Tropicale, EA 7292, Fédération de Médecine Translationelle (IPPTS), Université de Strasbourg, 3 rue Koeberlé, 67000 Strasbourg, France; ookutman@unistra.fr; 2Laboratoire de Diagnostic Génétique, UF3472-génétique de l’infertilité, Hôpitaux Universitaires de Strasbourg, 67000 Strasbourg, France; 3Memorial Atasehir Hospital, In Vitro Fertilization (IVF) Andrology and Genetics Center, Kucukbakkalkoy mh.Vedat Gunyol cd No:28-30, 34758 Atasehir/Istanbul, Turkey; cem.demirel@memorial.com.tr (C.D.); firat.tulek@memorial.com.tr (F.T.); 4Médecine Translationnelle et Neurogénétique, Institut de Génétique et de Biologie Moléculaire et Cellulaire (IGBMC), Institut National de Santé et de Recherche Médicale (INSERM), U964/Centre National de Recherche Scientifique (CNRS) UMR 7104, Université de Strasbourg, 67404 Illkirch, France; pfister@igbmc.fr (V.P.); ncharlet@igbmc.fr (N.C.-B.); 5Hibrigen Biotechnology R&D Industry and Trade Ltd. Co.,Tubitak MAM Teknoloji Serbest Bolgesi, Baris SB Mh 5002.sk Yeni Tek. Binasi ABlok 4, A/101 Gebze/Kocaeli, Turkey; umutbuyuk@hibrigen.com; 6Laboratoire de Diagnostic Génétique, Hôpitaux Universitaires de Strasbourg, 67000 Strasbourg, France; jeanmuller@unistra.fr; 7Laboratoire de Génétique Médicale, INSERM U1112, Fédération de Médecine Translationnelle de Strasbourg (FMTS), Université de Strasbourg, 67000 Strasbourg, France

**Keywords:** immature oocytes, oocyte maturation defect, female infertility

## Abstract

In vitro fertilization (IVF) involves controlled ovarian hyperstimulation using hormones to produce large numbers of oocytes. The success of IVF is tightly linked to the availability of mature oocytes. In most cases, about 70% to 80% of the oocytes are mature at the time of retrieval, however, in rare instances, all of them may be immature, implying that they were not able to reach the metaphase II (MII) stage. The failure to obtain any mature oocytes, despite a well conducted ovarian stimulation in repeated cycles is a very rare cause of primary female infertility, for which the underlying suspected genetic factors are still largely unknown. In this study, we present the whole exome sequencing analysis of a consanguineous Turkish family comprising three sisters with a recurrent oocyte maturation defect. Analysis of the data reveals a homozygous splice site mutation (c.1775-3C>A) in the zona pellucida glycoprotein 1 (*ZP1*) gene. Minigene experiments show that the mutation causes the retention of the intron 11 sequence between exon 11 and exon 12, resulting in a frameshift and the likely production of a truncated protein.

## 1. Introduction

Successful human fertilization requires the fusion of a sperm cell with a mature oocyte. Naturally, oogenesis involves two developmental arrests of the maturing oocytes, the first one at meiosis I, more precisely at the diplotene stage of prophase I, at which point they are named germinal vesicles (GV) and the second one at meiosis II, when they are named MII oocytes and considered mature oocytes. Upon a surge in luteinizing (LH) hormone, GV oocytes resume meiosis and exit prophase I [1], this involves chromatin condensation and breakdown of the nuclear envelope [2]. Coupled with spindle formation and chromosome alignment, oocytes enter into metaphase I (when they are named MI oocytes) which is completed by the extrusion of the first polar body. Oocytes then enter into meiosis II, which will stop at the metaphase II stage, fertilization will resume meiosis II and the second polar body will then be extruded [3].

Oocyte maturation arrest due to failure of germinal-vesicle breakdown (germinal-vesicle arrest), because of the absence of the first polar body extrusion (meiosis I arrest) or due to a failure to progress beyond metaphase II (metaphase II arrest) are rare causes of primary female infertility [4]. Genetic factors related to human oocyte maturation arrest have been poorly investigated and remain still largely unknown. According to the Online Mendelian Inheritance in Man (OMIM) database, mutations in seven genes have been correlated to oocyte maturation defects causing female infertility [5,6,7,8,9,10,11,12,13,14,15,16,17,18,19,20,21,22,23,24,25,26,27,28,29,30,31]. Among them mutations in *TUBB8*, *PATL2* and *WEE2* cause variable phenotypic deficiencies in oocyte maturation; including oocyte maturation arrest in the GV or MI stage, MII oocytes that cannot be fertilized or MII oocytes that can be fertilized but arrest at an early embryonic developmental stage [5,6,7,8,9,10,11,12,13,14,15,16,17,18,19,20,21]. 

By studying, via whole exome sequencing (WES), a consanguineous Turkish family comprising three sisters suffering from infertility due to repeated failures to obtain mature oocytes during IVF cycles in different centers, we identified a homozygous splice site mutation of the Zona Pellucida Glycoprotein 1 (*ZP1*, MIM195000). Homozygous or compound heterozygous mutations in *ZP1* have been identified before in women with primary infertility due to the absence of the oocyte zona pellucida and in female infertility associated with empty follicle syndrome or degenerate oocytes. Our findings expand the genetic spectrum of oocyte maturation defects caused by *ZP1* mutations. 

## 2. Materials and Methods 

### 2.1. Patients and Collected Samples

The family studied comprised three sisters with recurrent oocyte maturation defects, in which collected oocytes did not reach the MII stage (Figure 1). Parents are third degree cousins. All three sisters have a normal karyotype. Sisters VI:3 and VI:4 underwent one ovarian stimulation, where 8 and 11 immature oocytes were collected respectively. No maturation could be achieved even after extended culture. The third sister (VI:2) underwent seven stimulation cycles; during the first four cycles either no oocytes were collected or collected oocytes were immature. During the fifth cycle, 7 oocytes were collected, two were indicated as MII, ICSI was performed and 2 embryos were transferred without a successful pregnancy. During the 6^th^ cycle, following an uneventful stimulation with an appropriate peak of estradiol (E2) and progesterone (P4) hormone levels, 10 follicles were aspirated, but no oocytes were retrieved. During the seventh cycle, 8 oocytes were collected however all were immature (Table 1).

Blood samples from all siblings, parents and available non-affected family members (7 samples in total) were collected (Figure 1) after obtaining signed consent forms. Genomic DNA was extracted from peripheral blood using Hibrigen DNA Extraction-Blood kit (Hibrigen, Istanbul, Turkey), according to the manufacturer’s instructions. DNA samples were resuspended in water and quantified by nanodrop.

All exon and intron/exon boundaries of the two identified genes at the beginning of the project, namely the *TUBB8* and *PATL2* genes, were amplified and sequenced with the Sanger method in patient VI:4.

Ethical approval was given by the Comité de Protection de la Personne (CPP) of Strasbourg University Hospital, France (CPP 09/40—WAC-2008-438 1W DC-2009-I 002).

### 2.2. Whole Exome Sequencing and Data Analysis

Whole exome sequencing of patients VI:2, VI:4 and V:3 (Figure 1) was performed by Integragen Genomics (Integragen S.A., Evry, France). Briefly, 150 ng genomic DNA was sheared by sonication and 150–200 bp fragments were subjected to library preparation. Libraries were prepared using the NEBNext® Ultra II kit (New England Biolabs®, Ipswich, MA, USA) according to the manufacturer’s instructions. DNA libraries were then enriched (Twist Human Core Exome Enrichment System—Twist Bioscience—and IntegraGen Custom) and sequenced with Illumina HiSeq4000 with read length 2 × 75 bp following Illumina’s instructions. Image analysis and base calling is performed using Illumina Real Time Analysis (2.7.7) with default parameters.

Detected variants were also scored and ranked by VaRank [32], which provides all variations from the most to the least pathogenic in the sequenced area. Data processing and analysis were performed as described previously with minor changes [33]. Barcode application has allowed a quick overview of the presence/absence status of each variant and their zygosity status within the analyzed individuals; it was used for initial filtering of variants. Because of the loop of consanguinity in the family, we focused our first analysis on homozygous variants shared by both infertile sisters through the whole exome by assuming a recessive mode of inheritance. However, we did not exclude possible dominant or X-linked transmission; after a first round of analysis, we also checked for heterozygous and hemizygous variants. Further filtering was applied according to frequency (filtering out when >1%) in the largest available databases; Genome Aggregation Database (gnomAD) (http://gnomad.broadinstitute.org/) and 1000 Genomes Project (1000 g) (https://www.internationalgenome.org/). SIFT and PolyPhen2 have been used for prediction of pathogenicity of the missense mutations. Expression, possible suggested role in oocyte maturation, frequency data and an available KO model helped us to grade variations that pass our filtering process for further analysis.

### 2.3. Mutation Screening

Whole exome sequencing analysis of the selected variant and its segregation in the family were confirmed by polymerase chain reaction (PCR) and Sanger sequencing. Forward (cagctctgcccagtgtgata) and reverse (tgtttgctctgcctgaaatg) primers were designed via the Primer3Plus program and checked by SlicoPCR and PrimerBlast. They were utilized to amplify exon 12 of *ZP1* at 61.8 °C annealing temperature, by using Qiagen Multiplex PCR kit (Qiagen, Venlo, The Netherlands) in total 25 µL as a reaction volume and 35 cycles. 485 bp DNA amplicons were obtained, then purified and double-strand sequencing of each DNA fragment was performed by GATC Services, Eurofins Genomics (Ebersberg, Germany).

### 2.4. Minigene Assay

Minigenes encompassing human *ZP1* exon 11 to exon 12 with inclusion of the full 265 nts-long intron 11 from control or patient cases have been constructed. 

*ZP1* exon 11 to 12 wild type and mutant sequences were amplified from patient or control DNA respectively using the following primers aaaagcttGACAGCGACGATCCTCAGGTCACC (Fwd) and aaggatccTCAGTAGGTTTGAAATGATTGTTTATTGGG (Rev). PCR conditions were as follows: initial denaturation at 94 °C for 2 min; then denaturation at 94 °C for 50 s, annealing at 60 °C for 50 s and extension at 72 °C for 90 s for 27 cycles; finally extension at 72 °C for 5 min. PCR products were then cloned into pcDNA3.1+ upon Hind3 and BamH1 digestion. Chinese Hamster Ovary Cell (CHO) cells were cultured in Dulbecco’s modified Eagle Medium (DMEM F12), 5% fetal bovine serum and gentamicin at 37 °C in 5% CO_2_. Cells were plated in 6 well plates and transfected 24 h after plating in DMEM + 0.1% fetal bovine serum using FugenHD (Roche) according to the manufacturer’s protocol. 

Total RNA was isolated 24 h after transfection using acidic phenol (Trizol reagent, Life Technologies, Darmstadt, Germany) from transfected cells and cDNA synthesis reactions were carried out with Superscript II (Invitrogen, Carlsbad, California, USA) using random hexamers. PCRs were performed with 28 cycles of 45 s at 94 °C, 45 s at 62 °C and 45 s at 72 °C using the forward primer AGCGACGATCCTCAGGTCACCG located in *ZP1* exon 11 and the reverse primer ATTCACTGTCTGTTGCTTTCCC located within *ZP1* exon 12. PCR products were loaded and analyzed on a standard 2% TBE agarose gel.

## 3. Results

Over the nine cycles of ovarian hyper-stimulation performed for the three sisters, with the exception of one which provided two mature oocytes (sister VI:2), either no oocytes or only a few immature ones were obtained. All three sisters had their IVF cycles performed in different centers, therefore the data of all cycles had been gathered from the relevant embryology lab databases and confirmed by the embryologists in charge. During the only cycle that patient VI:4 underwent, 11 cumulus-oocyte complexes were retrieved; it has been noted that most of them seemed to have enlarged zona-pellucida (Figure 2A). However, oocyte denudation revealed that all oocytes were immature and zona-free (Figure 2B). Graphic illustration of follicular and endometrial growth during this patient’s ovarian stimulation is given on Figure 3.

Prior to WES, a Sanger sequencing analysis of *TUBB8* and *PATL2* genes was performed and revealed no mutation. 

In order to identify the genetic cause of oocyte maturation arrest in the family, WES was applied for samples V:3 (unaffected), VI:2 and VI:4 (affected) (Figure 1). For these samples, at least 5.8 GB DNA sequence were generated with >98.8% of the target exome being represented with >25-fold coverage (Table 2). WES analysis revealed a homozygous splice region variant in the two affected sisters on the *ZP1* gene (NM_207341.3), c.1775-3C>A. Its presence was confirmed in the third affected sister and segregation analysis of the available family members via Sanger sequencing (Figure 4C), showed that parents are heterozygous while V:3 and VI:1 are wild type for the substitution.

The c.1775-3C>A mutation affects the highly conserved CAG 3′ acceptor splice site of the last intron of *ZP1* (Figure 4A) and therefore may disrupt its splicing (Figure 5A) as suggested by the in silico strength splice predictors (−12% using Splice Site Finder [34] and −70.1% using MaxEntScan [35]). To confirm this hypothesis, we constructed a minigene from control or patient cases encompassing human *ZP1* exon 11 to exon 12 and thus including the full 265 nts-long intron 11 with the control (CAG) or mutant sequence (AAG) (Figure 5B). These minigene constructs were transfected into CHO cells and transcripts were analyzed by RT-PCR 24 h after transfection. As shown in Figure 5C, the c.1775-3C>A mutation leads to a complete retention of intron 11, which is spliced out from the wild type sequence. However, note that even with the control sequence intron 11 is not fully spliced and some unspliced product exists in CHO cells, likely due to the small size of this intron (265 nucleotides) and its weak 3′ splice site sequence that forms a hairpin impairing splicing (Figure 5D). 

## 4. Discussion

The zona pellucida (ZP) is an extracellular matrix that surrounds the oocyte and preimplantation embryo. The mammalian zona pellucida mediates species-specific sperm binding while induction of the acrosome reaction prevents post-fertilization polyspermy and protects the newly formed zygote during its journey down the fallopian tube. In humans there are four glycoproteins that form the zona pellucida: ZP1, ZP2, ZP3 and ZP4. ZP1 is the least abundant of the four human ZP glycoproteins, but it ensures the structural integrity of the zona pellucida. 

The *ZP1* gene encodes a 638 amino acid glycoprotein. It contains an N-terminal signal peptide (SP), a cysteine rich P-type Trefoil domain and three ZP domains (ZP-N1, ZP-N, ZP-C), a consensus furin cleavage site (CFCS) and a transmembrane domain (TMD) (Figure 4B). During the formation of the zona pellucida, ZP1 plays a critical role by cross-linking the long chains of filaments made of consecutive alternates of ZP2, ZP3 and ZP4 [28]. 

In humans, filament cross-linking by ZP1 is crucial to form a stable ZP. The cross-linking function of ZP1 maps to its N-terminal ZP-N1 domain [36]. *Zp1* null female mice have an egg coat that is looser and fragile [37]. 

It is apparent that *ZP1* mutations can be detrimental either by completely abolishing ZP cross-linking due to impairment of protein secretion or by affecting ZP1 residues which play a structural role in cross-link formation [36]. Human homozygous or compound heterozygous mutations reported in *ZP1* result in similar albeit not identical phenotypes (Figure 2A, Table 3). Indeed, mutations in *ZP1* can lead to oocyte degeneration, oocytes lacking zona pellucida or increased fragility of oocytes rendering follicular puncture tricky, ultimately resulting in an empty follicle syndrome [23,24,25,26,27,28]

Among the mutations reported in the literature, seven of them (no 2,3,4,8,10,11,13 as indicated on Table 3) affect one of the zonae pellucidae domains (ZP-N1, ZP-N or ZP-C domain). These mutations result in a lack of oocytes or only degenerate oocytes. Other mutations are located between domains of the ZP1 protein and lead to the lack of oocytes or to zona-free oocytes.

So far, four splice site mutations have been reported on *ZP1*. Only zona-free oocytes were retrieved from women carrying compound heterozygous splice site mutation (c.1430 + 1G > T and c.1775-8T > C [25]. A homozygous splice site mutation (c.1014+1G > A) has been identified in a patient for whom several empty cumulus–oocyte complexes (COC) were retrieved [27]. Ovarian stimulation resulted in either COC or no oocytes for another patient in the same study with biallelic mutations in *ZP1*, namely [c.1573-2A > G and c.508del]. Among four splicing mutations, only c.1775-8T does not fall within one of the ZP-domains. It was shown for *ZP1* cDNA carrying variant c.1775-8T > C, that the 265-bp intronic sequence between exon 11 and exon 12 was retained and probably resulted in a frameshift and protein truncation [25]. 

Our study identified a homozygous splice site mutation (c.1775-3C>A) in the *ZP1* gene in three sisters from a consanguineous Turkish family. The mutation is located just before the last exon. A minigene assay revealed that the mutation causes intron retention in affected patients. Interestingly, we noted a partial retention of intron 11 even with the WT sequence. This incomplete splicing of the WT sequence in CHO cells is most probably due to the fact that this intron is short and that its 3′ splice site sequence is weak. Failure to remove intron 11 is predicted to result in a shorter ZP1 protein which contains all domains but is devoid of its transmembrane domain located in the last exon 12. The TMD is essential for anchoring the zona proteins on the plasma membrane. Previous studies have shown that the ZP2 and ZP3 proteins truncated of their transmembrane domain were unable to incorporate into the zona pellucida [38] The absence of the ZP1 transmembrane domain probably results in an inactive protein unable to assemble with ZP2 ZP3 and ZP4 to form a correct zona pellucida. In the case presented in this article, either the retention of intron 11 results in a misfolded protein because of the presence of a premature stop codon or a truncated ZP1 protein lacking the TMD may exist leading to abnormal zona formation hindering oocyte maturation.

Our results revealed a homozygous splice mutation in *ZP1* that was associated with immature zona-free oocytes and expanded the genetic spectrum of oocyte maturation defects caused by *ZP1* mutations.

## Figures and Tables

**Figure 1 genes-11-00382-f001:**
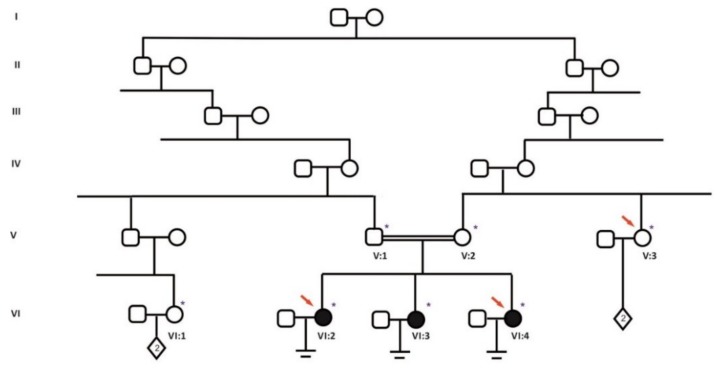
Consanguineous Turkish family with female infertility characterized by an oocyte maturation defect. Filled symbols indicate affected members and clear symbols indicate unaffected members. Blue asterisk (*****) indicates collected samples, red arrows show samples for which WES has been performed.

**Figure 2 genes-11-00382-f002:**
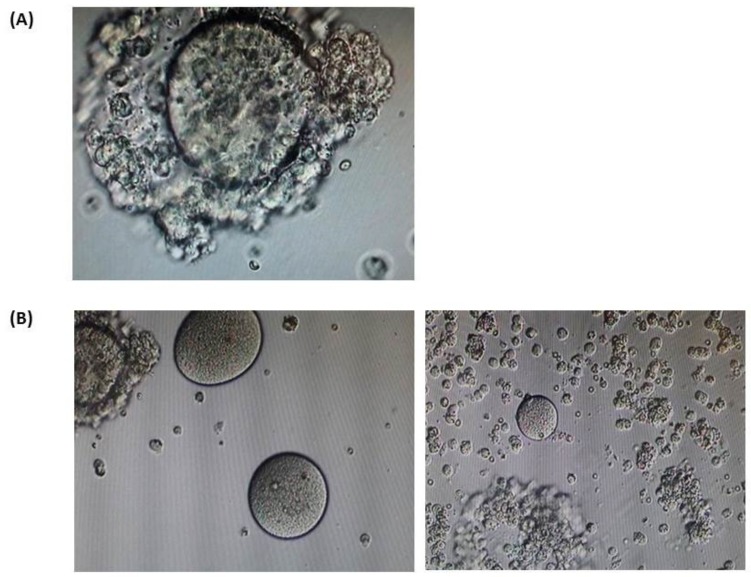
Oocytes from patient VI:4. (**A**) An example of cumulus-oocyte complex (COC) from patient IV:4. (**B**) Denuded oocytes. Only zona-free, immature oocytes were retrieved.

**Figure 3 genes-11-00382-f003:**
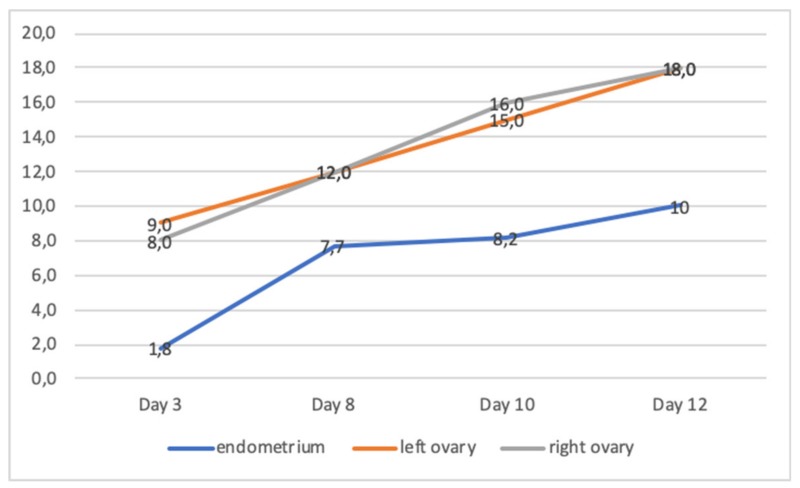
Follicular and endometrial growth (y axis in mm) during controlled ovarian stimulation of patient VI:4, where 11 immature oocytes had been collected with defective zona pellucidae. Leading follicle sizes were plotted for right and left ovaries. Blood estradiol level was 6580 pg/mL on day 12 and ovulation trigger was obtained using 0.2 mg triptorelin acetate.

**Figure 4 genes-11-00382-f004:**
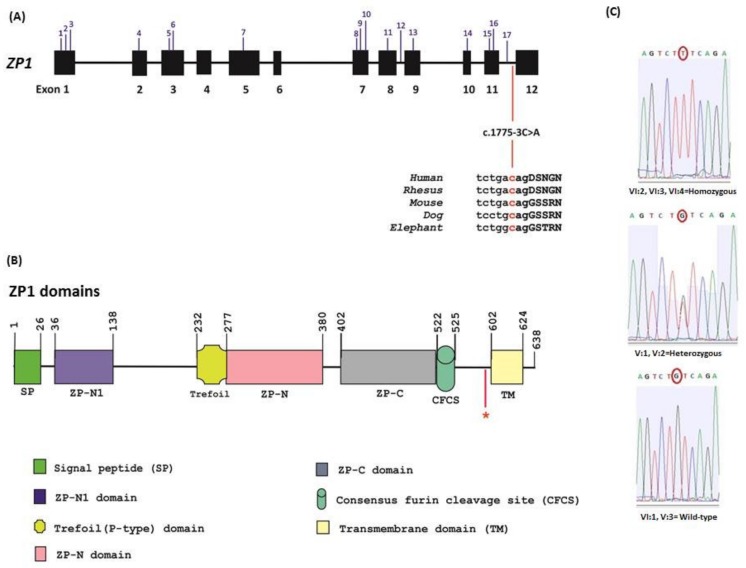
Splice site mutation in Turkish family with oocyte maturation defect phenotype. (**A**) Schematic diagram of *ZP1* exons is shown. The mutations identified so far on *ZP1* are indicated in blue on top, numbered 1 to 17 [23,24,25,26,27,28], details are given in Table 3. The bottom part indicates the newly identified nucleotide change within a sequence alignment among mammals. (**B**) Domain architecture of wild-type ZP1 (adopted from Nishimura et al. 2019, [36]). Domain boundaries are indicated and the identified mutation in this study is marked with a red asterisk. (**C**) Plots of results from Sanger sequencing for mutant, carrier and wild-type samples. The new allele identified is highlighted with a red circle.

**Figure 5 genes-11-00382-f005:**
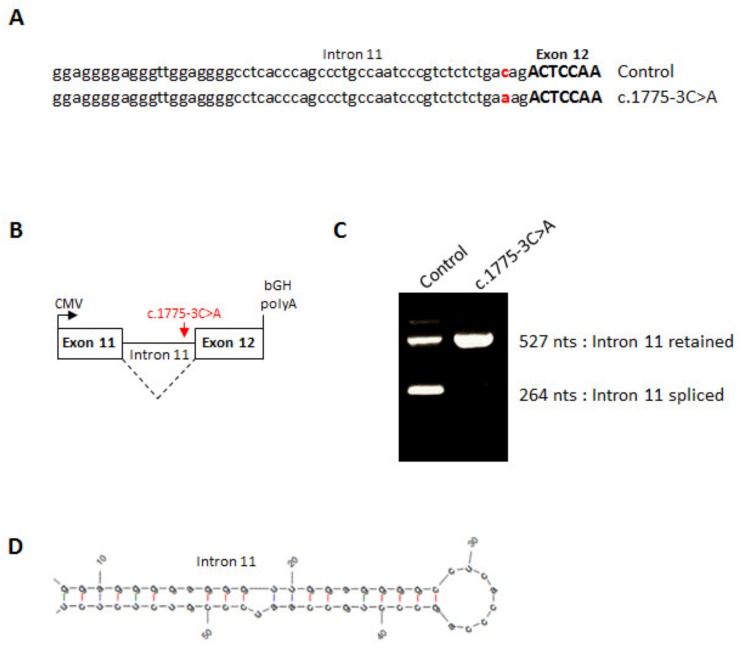
Splicing effect of the *ZP1* mutation. (**A**) Sequence of human *ZP1* from the end of intron 11 up to the start of exon 12 from a control minigene or a c.1775-3C>A minigene. (**B**) Line drawing of *ZP1* minigene cloned into pcDNA3.1. (**C**) RT-PCR of wild type or mutant *ZP1* mRNA expressed in CHO cells. (**D**) RNA hairpin structure formed by the end of *ZP1* intron 11.

**Table 1 genes-11-00382-t001:** Clinical data from three affected sisters. MII: mature oocytes, metaphase II, ET: embryo transfer.

Patient	Age	Karyotype	Cycle	Collected oocytes	Results
VI:2	38	46,XX	1-4	No oocyte OR no MII	No ET, cancelled
			5	7 oocytes, 2xMII	ET, no pregnancy
			6	10 follicles were aspirated, no oocytes	No ET, cancelled
			7	8 oocytes, no MII	No ET, cancelled
VI:3	36	46,XX	1	8 oocytes, no MII	No ET, cancelled
VI:4	28	46,XX	1	11 oocytes, no MII	No ET, cancelled

**Table 2 genes-11-00382-t002:** Yield of whole exome sequencing of 3 samples.

Sample ID	Yield-GB	%Align	Coverage above X25
VI:2	6.125	98.85	99.6
VI:4	5.859	98.91	99.5
V:3	6.913	99.6	98.88

**Table 3 genes-11-00382-t003:** Identified *ZP1* mutations described in the literature with corresponding zygosity and phenotype. aa change: amino acid change in the protein. (*): in this study one patient had both c.247T>C and c.1413G>A, however no segregation study has been done; the second one was reported as heterozygous for c.1413G>A, which is the only heterozygous *ZP1* mutation reported in the literature for oocyte maturation defects. No 7, 14, 12 and 17 were splice site variants; they were predicted to abolish the donor or acceptor splice sites of intron 5, intron 9, intron 8 and intron 11 respectively.

No	cDNA	aa Change	Zygosity	Phenotype	Ref
1	c.123C>A	p.Tyr41*	Comp Het^(1)^	No oocyte	[27]
2	c.170-174del	p.Gly57Aspfs*9	Comp Het^(2)^	No oocyte	[28]
3	c.181C>T	p.Arg61Cys	Comp Het^(3)^	No oocyte	[26]
4	c.247T>C	p.Trp83Arg	Het*	Degenerate oocyte	[23]
5	c.507del	p.His170Ilefs*52	Hom	No oocyte	[25]
6	c.508del	p.His170Ile*52	Comp Het^(4)^	No oocyte	[27]
7	c.1014+1G>A	p.?	Hom	No oocyte	[27]
8	c.1129-1130del	p.Val377Leufs*5	Hom	No oocyte	[27]
9	c.1169-1176del	p.Ile390Thrfs*16	Comp Het^(2,3)^ Hom	No oocyte Zona free oocytes	[26,28] [24]
10	c.1228C>T	p.Arg410Trp	Hom	Degenerate oocyte	[25]
11	c.1413G>A	p.Trp471*	Het*	Degenerate oocyte	[23]
12	c.1430+1G>T	p.?	Comp Het^(5)^	Zona free oocytes	[25]
13	c.1510C>T	p.Arg504*	Hom	No oocyte	[27]
14	c.1573-2A>G	p.?	Comp Het^(4)^	No oocyte	[27]
15	c.1663C>T	p.Arg555*	Comp Het^(1)^	No oocyte	[27]
16	c.1708G>A	p.Val570Met	Hom	No oocyte or zona free oocytes	[25]
17	c.1775-8T>C	p.?	Comp Het^(5)^	Zona free oocytes	[25]

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
