# Peer review of "Homozygous Splice Site Mutation in ZP1 Causes Familial Oocyte Maturation Defect"

_genes, 2020, doi:10.3390/genes11040382_

Round 1
Reviewer 1 Report
This manuscript reports the finding that a point mutation (c.1775-3C>A) in intron 11 of the human zona pellucida glycoprotein 1 (ZP1) gene causes oocyte maturation defect and female infertility. This finding is novel and very interesting.
The manuscript is poor written and organized. Data analysis and discussion are not comprehensive.
Reviewer 2 Report
This manuscript describes a novel homozygous splice junction mutation in the ZP1 gene in three sisters with infertility ascribed to oocyte maturation failure. The genetic analysis is sound. Although this is not the first report of ZP1 mutations causing female infertility, the phenotype of the affected women is different from previously reported cases. The work would be improved if the authors are able to include more information on the characteristics of follicle growth and the morphology of aspirated oocytes.
In Table 1, subject age not birth date should be reported. The hormone levels presented are not informative and should be removed or presented as supplemental data. If the authors could put the aspirated follicle sizes in Table 1 it would be more helpful. The authors should include a description of the oocytes aspirated, including the morphology of the zona pellucida and cumulus complex in the Results section (Table 1), not the Discussion. Are images of representative aspirated oocytes available? It is not clear whether IRB approval for this study was required and if the study was submitted for IRB approval it should be stated. An expanded discussion of the structure/function relationships based on known ZP1 mutations and phenotypes is warranted. The manuscript needs copy-editing to improve presentation/English usage.
